# Spatial Allocation Based on Physiological Needs and Land Suitability Using the Combination of Ecological Footprint and SVM (Case Study: Java Island, Indonesia)

Sitarani Safitri [1,2,*], Ketut Wikantika [1], Akhmad Riqqi [1,2], Albertus Deliar [1] and Irawan Sumarto [1]

1 Remote Sensing and Geographic Information Science Research Group, Department of Geodesy and Geomatics, Faculty of Earth Science and Technology, Bandung Institute of Technology, Bandung 40132, Indonesia; ketut@gd.itb.ac.id (K.W.); riqqi@gd.itb.ac.id (A.R.); albert@gd.itb.ac.id (A.D.); irawan@gd.itb.ac.id (I.S.)
2 Lokahita, Research Center for Sustainable Ecology and Geospatial, Bandung 40265, Indonesia
* Correspondence: sitaranisafitri@yayasanlokahita.org

**Abstract:** Indonesia currently has 269 million people or 3.49% of the world's total population and is ranked as the fourth most populous country in the world. Analysis by the Ministry of Public Works and Public Housing of Indonesia in 2010 shows that Java's biocapacity is already experiencing a deficit. Therefore, optimization needs to be done to reduce deficits. This study aims to optimize and assess spatial allocation accuracy based on land-use/land cover suitability. In this study, the ecological footprint (EF) is utilized as a spatial allocation assessment based on physiological needs. The concept of land suitability aims for optimal and sustainable land use. Moreover, the land suitability model was conducted using the support vector machine (SVM). SVM is used to find the best hyperplane by maximizing the distance between classes. A hyperplane is a function that can be used to separate land-use/land cover types. The land suitability model's overall-accuracy model was 86.46%, with a kappa coefficient value of 0.812. The final results show that agricultural land, plantations, and pastureland are still experiencing deficits, but there is some reduction. The deficit reduction for agricultural land reached 510,588.49 ha, 18,986.14 ha for plantations, and 1015.94 ha for pastures. The results indicate that the SVM algorithm is efficient in mapping the land-use suitability and optimizing spatial allocation.

**Keywords:** spatial allocation; land suitability; ecological footprint; SVM (support vector machine)

## 1. Introduction

The land can produce products in the form of goods and services (supply) to meet human needs (demand). The needs in question include food, water, air, homeostasis, rest, and excretion as the most basic human physiological needs that must be met based on Maslow's Hierarchy of Needs [1–4]. Products produced by land include food and fiber, as well as environmental services such as water supply, water flow, and air quality control. The land's ability to make these products depends on ecological quality [5,6]. The decline in ecological quality is influenced by environmental pressures due to ecosystem changes to meet human needs, such as changes in land cover, resource retrieval and depletion (such as logging and overfishing), disposal and pollution of emissions, and modification and movement of organisms [7,8]. The resulting environmental impacts include, but are not limited to, climate change, land degradation, loss of biodiversity, and environmental pollution [9–11]. These ecological problems reduce land productivity, which has become a global issue in recent decades [3,4]. Therefore, it is necessary to plan for sustainable development in meeting human needs.

The fulfillment of human needs through development activities requires the allocation of land. The spatial allocation represents the process of determining the amount of land for

specific uses (or unused) through legal and administrative steps, which leads to the implementation of planning [12]. Thus, spatial allocation determines economic development performance and environmental quality. Land in spatial allocation terminology refers to the terms of land cover and land use. Land cover is the various biophysical materials found on the land, such as buildings, vegetation, and roads. Land use has a different definition characterized by regulating and using multiple land-cover types [13,14]. This definition of land use describes the direct relationship between land cover and human activities. Along with developing the understanding of land-cover and land-use, there are still differences in opinion regarding the classification division. Therefore, land cover and land use are often used in a unified term, even though they have different definitions [15–17]. This study then uses the term land-use/land cover.

Several methods have been developed to calculate spatial allocation. One of the methods referred to is the life cycle analysis (LCA), a tool to assess and measure the total environmental impact based on the entire life cycle of a good or service, including raw materials, processes, products, and technology or relevant activities [18,19]. For example, rice is assessed for its environmental impact from planting rice seeds, cultivation and maintenance processes (irrigation, pesticides), harvesting, milling grains into rice grains, processing food from rice, consumption, and waste disposal. However, this approach is too particular, so that each commodity will have a different valuation process. Another approach that has been widely used is the ecological footprint (EF). This approach uses land-based indicators to assess resource sustainability, i.e., by comparing land needs and availability to meet the needs of specific populations [5,6]. EF has transparent metric units of calculation, generally available data needs, and standardized measurement methods [20–22]. Based on existing developments, EF becomes a comprehensive, but straightforward spatial allocation calculation method. Therefore, the EF approach is used in this study to calculate the spatial allocation.

The availability of productive land and resource production each year is limited. Therefore, the measurement of various ecosystem services and ecological resources uses metric units of the area [23]. Galli et al. (2015) explained that there are three types of unit area in the calculation of ecological footprint, namely, global hectares (gha), world-average hectares (wha), and nation-specific actual hectares (ha) [24]. One global hectare (gha) is equivalent to one hectare with the world average biological productivity of all land use/land cover for a given year. World-average hectares (wha) is the area of a specific land use/land cover (e.g., the area of wetland agriculture) with its world-average productivity (e.g., the world-average productivity of wetland agriculture). Finally, actual hectares (ha) are the physical area of a particular type of land use/land cover located in a particular area characterized by productivity in that area. Actual hectares (ha) are useful for visualizing the physical land area occupied for a particular activity. The calculation of the ecological footprint includes six types/classes of productive land to meet human needs, namely, cropland, grazing land, forest, fisheries, energy land, and built-up land [23,25]. Each type of land use/land cover has a specific annual production amount, which can help meet human needs.

The ecological footprint approach is very sensitive to the spatial-temporal scale used. This nature is due to the variation in the resources generated and needed in different countries or regions [26–31]. Most research on ecological footprint uses units of global hectares (gha). Several studies use actual hectares (ha) to suit the country or region circumstances [32–37]. The ecological footprint approach using generalized gha and wha metric units cannot represent the difference between local land supply and demand. Meanwhile, the ecological footprint assessment using actual hectares (ha) can calculate the land demand locally.

The spatial allocation of land use/land cover in this study was carried out using an ecological footprint approach, as done by Lane et al. (2014) [38,39]. This model utilizes actual hectares (ha) because the spatial allocation is specific for the study area according to land productivity in Java Island and is specific for each type of land use/land cover.

The land use/land cover types allocated were adopted from the classification of six main types of productive land by the World Conservation Union [40]. Notably, in this study, the land use/land cover is classified into eight types/classes by classifying agricultural land (cropland) to be more specific, namely, wetland agriculture (rice fields), dryland agriculture, and plantation.

Furthermore, the calculation of spatial allocation certainly needs to pay attention to its suitability. The concept of land suitability aims for optimal and sustainable land use [41]. The aim is to identify the most appropriate spatial patterns for future land use [42,43]. Therefore, land suitability analysis is an essential part of urban planning and management. The land characteristics can be used to assess land suitability, namely land attributes that can be measured or estimated, such as slope, rainfall, soil texture, and vegetation [41]. The land characteristics in the assessment system are assumed to be able to determine the direction of spatial allocation. More specifically, land suitability assessments pay attention to the interactions between land characteristics. For example, land suitable for paddy fields is not determined by the angle of slope alone, but by the interaction between slope angle, slope length, permeability, soil type, rainfall intensity, and other characteristics. Owing to these interaction problems, it is recommended that land use/land cover suitability assessment should be carried out in terms of land quality. The quality of land formation is one of the most influential parameters on the quality of land suitability assessment and the reliability of land use plans [44]. Soil quality is an intricate attribute of soil containing one or more soil characteristics. In recent decades, land suitability analysis has been applied to agricultural land assessments [45], determination of land as habitat for various species of flora and fauna [46,47], landscape evaluation and planning [48,49], along with regional planning and environmental impact assessments [50,51].

In this study, the land suitability model was conducted using support vector machine (SVM). SVM is used to find the best hyperplane by maximizing the distance between classes. A hyperplane is a function that can be used to separate classes [52]. SVM separates land suitability classes based on the land characteristics used as parameters. Although SVM has been widely used for land mapping in recent years, only a few have used SVM to map land suitability [53–56]. The use of SVM for land suitability in previous studies also focused more on agricultural land, with accuracy and kappa reaching more than 75% or equivalent to a very high level of suitability between the model and existing conditions. The results of these studies are taken into consideration to choose SVM. The novelty that distinguishes this study from previous research is the utilization of SVM to assess land suitability for seven land use/land cover classes. The final result of the classification provides delineation information on land suitability so that the fulfillment of spatial allocation can be calculated. The research results are expected to provide an overview of policymaking related to land use planning. Therefore, there are three specific objectives: (1) calculating land demand based on physiological needs using ecological footprint (EF), (2) assessing land suitability for seven land use/land cover using SVM, and (3) calculating the possible fulfillment for spatial allocation.

## 2. Materials and Methods

### 2.1. Materials

The study was conducted in Java Island, Indonesia, which is 129,438.28 km$^2$ (Figure 1). This island is inhabited by more than 149 million people, making it the most populated island in Indonesia [57]. The population growth rate in Java Island in 2018 reached 1.23%, with a population density of 1317 people/km$^2$, and 56.7% of the total population lives in urban areas [58]. The population of Java Island is almost the same as the total population of other islands in Indonesia. Energy consumption in Indonesia is focused on Java Island, or more than 60% of the total national consumption, because 57% of its consumers are in Java [59]. Based on the evaluation of the ecological footprint conducted by the Ministry of Public Works of the Republic of Indonesia [37], the biocapacity of Java island has an overall deficit. This condition is suitable for testing spatial allocation models. Biocapacity is the

land's ability to generate natural resources, and absorb and filter other materials such as carbon dioxide from the atmosphere. The selection of this study area will provide a more in-depth understanding related to spatial allocation and sustainability.

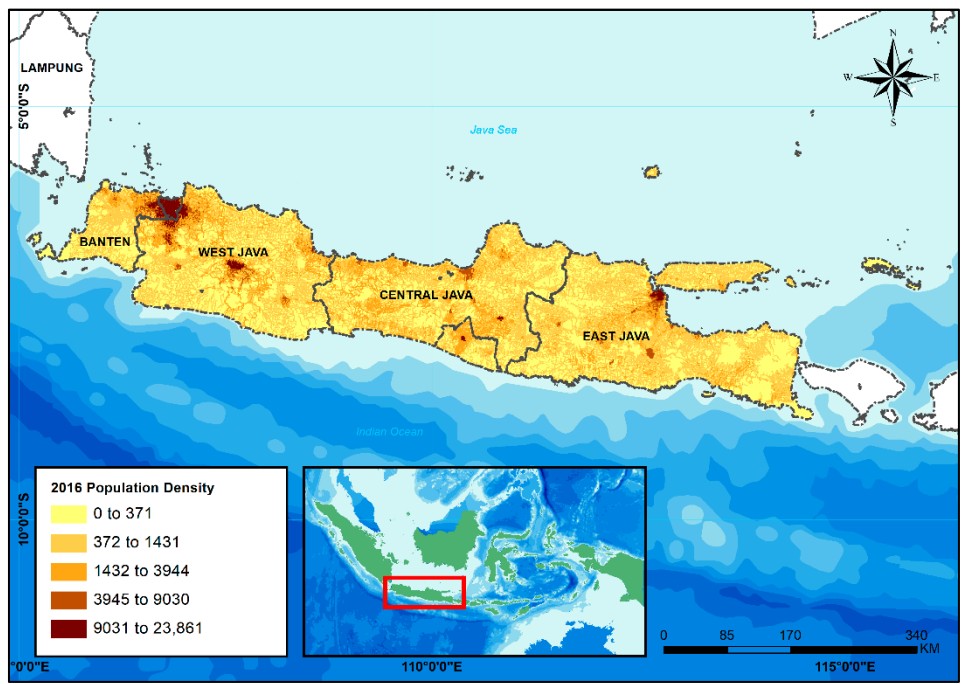

**Figure 1.** Population distribution of Java Island in the year 2016.

The data collected in this study are secondary data identified based on the variables in the hypothesis. The data are generally obtained from government agencies and research institutes. Table 1 shows the information related to data types and data sources. According to Maslow's Theory, the basic needs to calculate spatial allocation include food, clothing, housing, public space, and energy, while the land suitability model is conducted using land characteristics as parameters. Land characteristics, according to the Food and Agriculture Organization (FAO), are land attributes that can be measured or estimated. The chosen land characteristics are altitude, slope, ecoregion, land surface temperature, rainfall, soil type, soil pH, water availability, and soil organic content.

Some of the secondary data that were collected in this study were pre-processed. Elevation and slope were extracted from Shuttle Radar Topography Mission (SRTM) data data. Land surface temperature (LST) was retrieved using bands 4, 5, and 10 with an algorithm created in ERDAS IMAGINE 2014 [60]. Land characteristics were selected as land suitability parameters based on literature studies and significance tests. The results of the Kolmogorov–Smirnov statistical test showed a significance value between 0 and 0.024 for all parameters. These significance values indicate the level of confidence in the correlation hypothesis between parameters and land-use/land cover to be rejected between 0% and 0.024% (significance <0.05%). Thus, for the 95% confidence level, the hypothesis that all parameters have a relationship with land-use/land cover is accepted. These land characteristics can determine suitable land-use/land cover. The spatial unit used in deciding the land-use/land cover location is a grid with a resolution of $30'' \times 30''$ ($\approx 0.9$ km $\times$ 0.9 km). Each grid has the values of these nine parameters. Normalization was conducted on the land suitability parameters (value range 0–1) by the min–max normalization method. Filling in each parameter's value into each grid was done using the maximum combined area (MCA) method. The MCA principle, namely the type or value of polygons (can be more than 1) in each grid with the largest total area considered dominant, will be the grid's value. The visualization of the land suitability parameters can be seen in Figures 2–5.

**Table 1.** Secondary data to calculate spatial allocation based on physiological needs and land suitability.

| Parameter/Data | Source(s) |
|---|---|
| Land cover (1:250,000, in 2016) | Ministry of Environment and Forestry—Kementerian Lingkungan Hidup dan Kehutanan (KLHK), Indonesia |
| Province statistics in Java (in 2016) | Central Bureau of Statistics, Indonesia |
| Agriculture Statistics (in 2016) | Ministry of Agriculture, Indonesia |
| Animal Husbandry and Health Statistics (in 2016) | |
| Electrical Statistics (2016) | Ministry of Energy and Mineral Resources, Indonesia |
| Elevation (resolution 90 m, in 2016) | Shuttle Radar Topography Mission (SRTM) data from NASA, provided by USGS Earth Resources Observation and Science (EROS) Data Center |
| Slope (resolution 90 m, in 2016) | |
| Ekoregion (1:500,000, in 2017) | Ministry of Environment and Forestry, Indonesia |
| Land surface temperature (resolution 90 m, in 2016) | Landsat 8 OLI from NASA, provided by USGS EROS Data Center |
| Rainfall | Central Bureau of Statistics, Indonesia |
| Soil type, soil pH, soil organic content (resolution 90 m) | International Soil Reference and Information Centre (ISRIC)—World Soil Information, Wageningen University and Research (WUR) |
| Water availability (per WD, in 2016) | Ministry of Public Works, Indonesia |

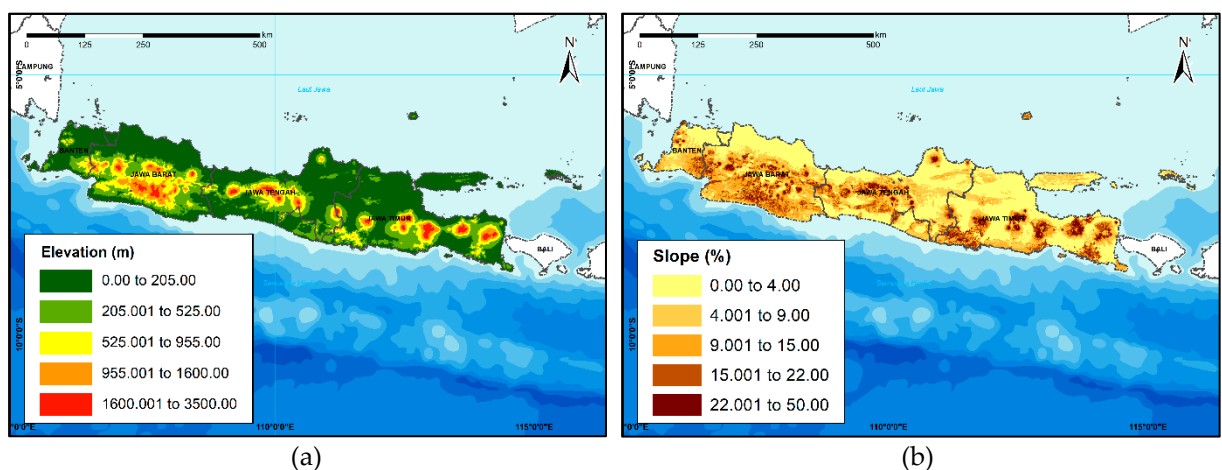

(a)    (b)

**Figure 2.** The topography parameters: elevation (**a**) and slope (**b**).

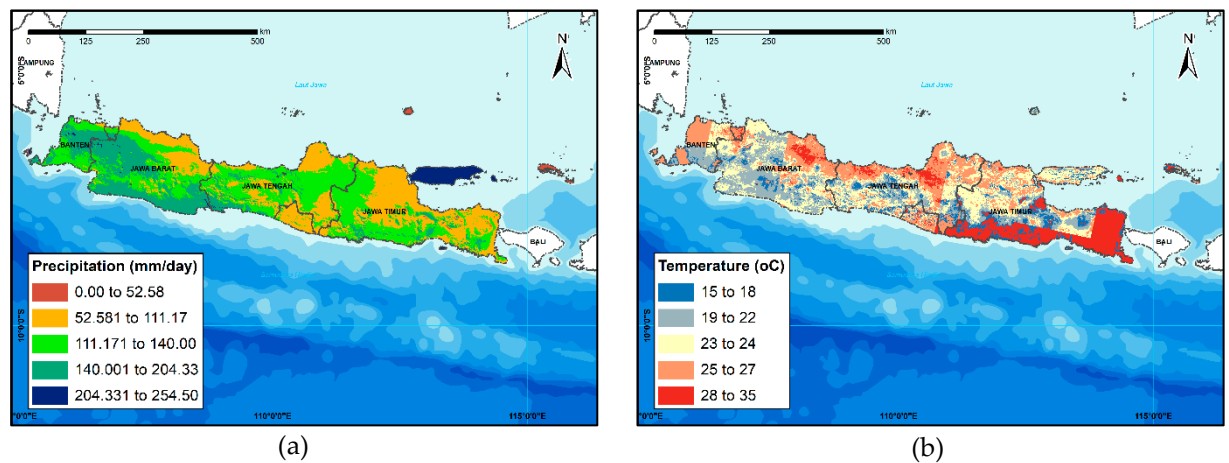

(a)    (b)

**Figure 3.** The climate parameters: rainfall/precipitation (**a**) and temperature (**b**).

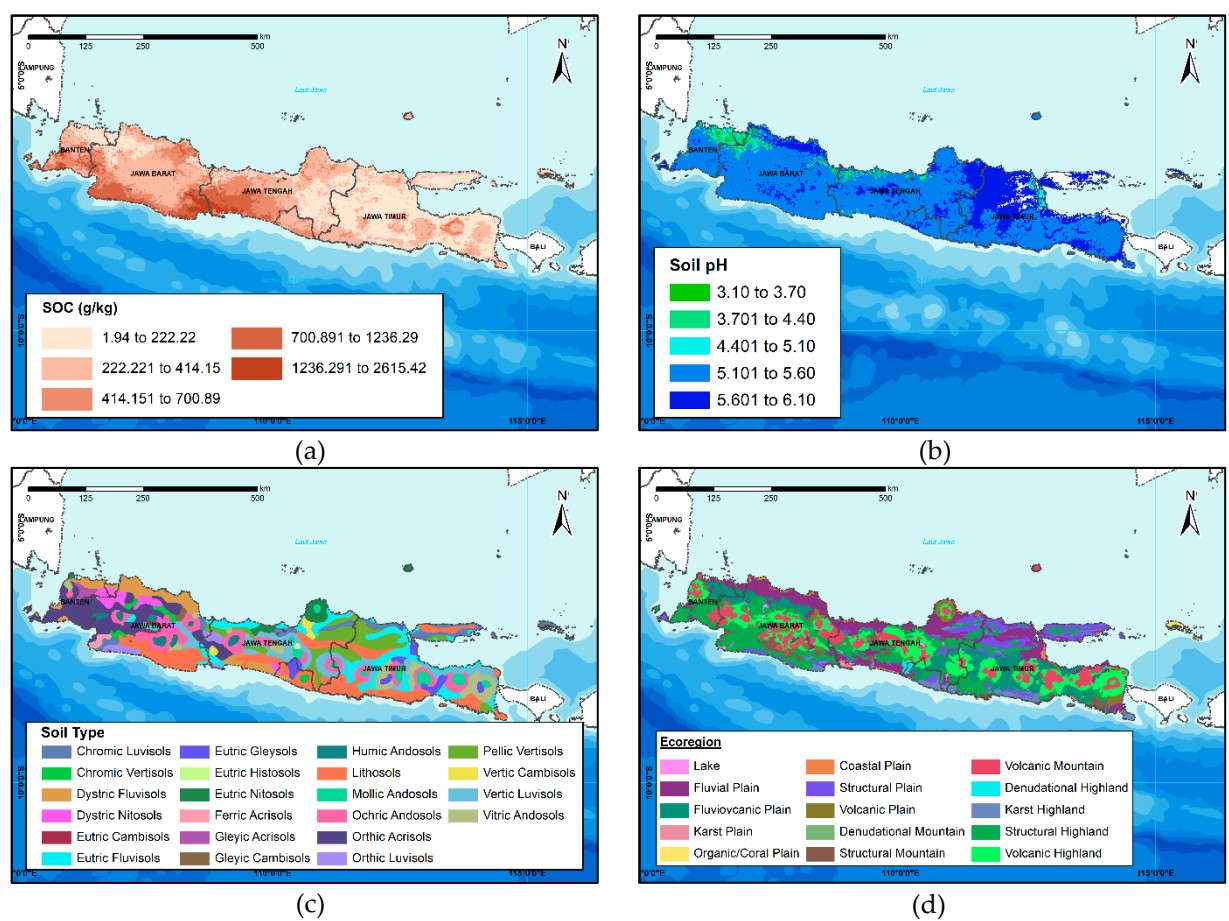

**Figure 4.** The soil characteristics: soil organic carbon (SOC) (**a**), soil pH (**b**), soil type (**c**), and ecoregion (**d**).

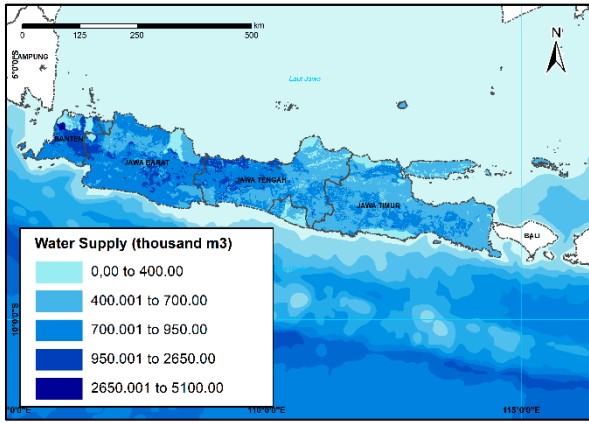

**Figure 5.** The water supply.

### 2.2. Methods

This research's nature is descriptive-quantitative, which describes the relationship between phenomena systematically, factually, and accurately. This study aims to determine the spatial allocation and suitability of land use/land cover deductively. In order to solve the research problem, several stages must be carried out in Figure 6. These stages generally consist of the following:

1. Perform the calculation of land use/land cover spatial allocation based on physiological needs using an ecological footprint approach with land use/land cover data and statistical data. The spatial allocation can also be carried out using several scenarios of meeting the needs.

2. Conduct land suitability analysis using the SVM with kernel trick. There are nine parameters and several sample points. The number of sample points and the sampling method refer to the standards set for geospatial information, namely SNI ISO 19157.

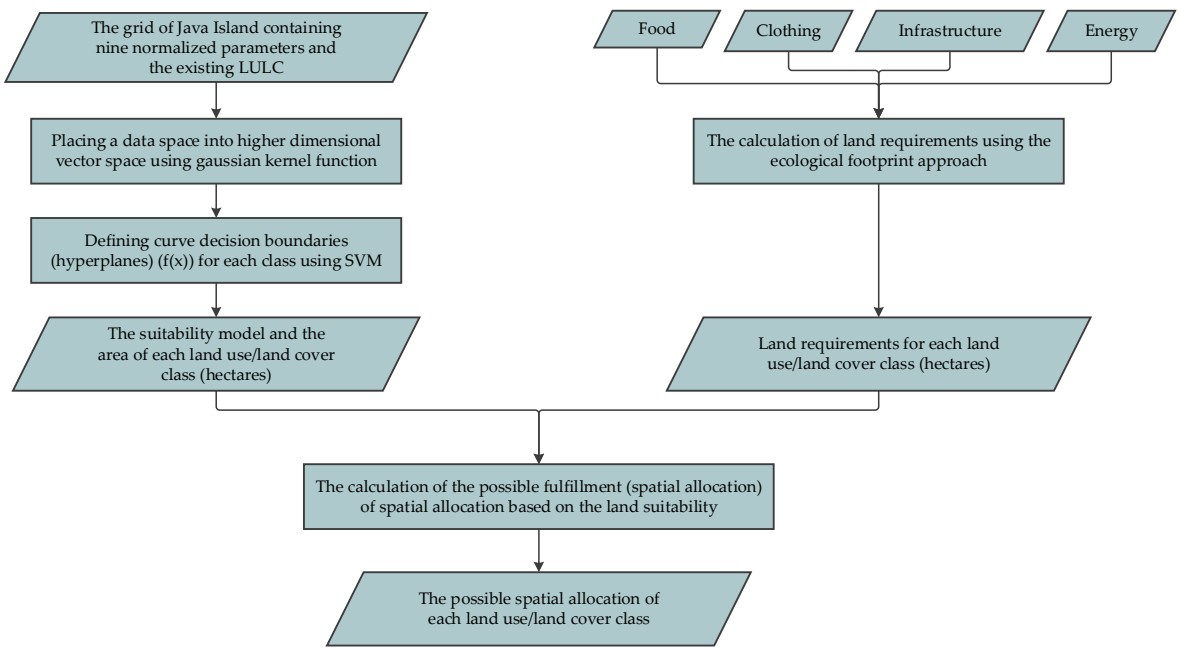

**Figure 6.** The workflow to calculate spatial allocation based on physiological needs and land suitability using the combination of ecological footprint and support vector machine (SVM). LULC, land-use/land cover.

### 2.2.1. The Calculation of Spatial Allocation with the Ecological Footprint (EF) Approach

A widely used approach developed to calculate spatial allocation is the ecological footprint (EF) approach. This approach uses land-based indicators to assess resource sustainability, i.e., by comparing land needs and availability to meet the needs of specific populations [5,61]. EF can provide information on the long-term ecological status and early warning for potential ecological risks. Furthermore, EF advantages include transparent metric units of calculation, generally available data needs, and standardized measurement methods [22,62]. Based on existing developments, EF is a simple, but comprehensive method of environmental sustainability.

The EF concept discusses how to reduce the population's impact on nature in at least two ways [63]. First, the ecological footprint measures the total ecological cost (within the land area) of the supply of all goods and services to the population. This cost shows that residents directly need land to produce agriculture, roads, buildings, and others. Still, indirectly, the land also contributes to the realizing goods and services consumed by the population. In this way, ecological footprints can be used to structure the ecological costs of population activity. Second, the ecological footprint is an indicator of sustainability, namely, environmental carrying capacity, which is the maximum population supported by a specific land area. This concept refers to all members of the ecosystem.

According to Maslow's Theory, there are some basic human needs: food, clothing, shelter and public space, and energy. Ecological footprint (EF) capable of quantifying land requirements to meet human needs [6]. Land requirements using the EF approach are calculated based on the amount of population consumption per capita. Each type of land cover/land use can provide specific resources. Therefore, calculating each land use/land cover's needs is explicitly carried out for every basic need. According to Wackernagel et al.

(2019), the equations for calculating ecological footprints in global hectares (gha) or global m$^2$ (gm$^2$) units are as follows:

$$EF = \sum_i \frac{P_i}{Y_{N,i}} \times YF_{N,i} \times EQF_i = P_i \times FI_i \times YF_{N,i} \times EQF_i \qquad (1)$$

where

| | | |
|---|---|---|
| EF | = | ecological footprint or land requirements (gm$^2$); |
| P | = | number of basic human needs (kg); |
| Y | = | productivity $\left(\text{kg/m}^2\right) = \frac{1}{\text{FI}}$; |
| FI | = | footprint intensity (m$^2$/kg); |
| YF | = | yield factor (wm$^2$/m$^2$); |
| EQF | = | equivalence factor (gm$^2$/wm$^2$). |

Pre-processing is carried out on several variables in Equation (1). The amount of production divided by the harvested area gives the productivity value. The productivity value is inversely related to the intensity footprint. YF and EQF values for the Indonesian region are available for download on the Ecological Footprint Explorer—http://data.footprintnetwork.org (accessed on 10 March 2021) by the Global Footprint Network [25,64].

### 2.2.2. Land Suitability Model with Support Vector Machine (SVM)

Land suitability assessment is an essential part of urban planning and management. The goal is to identify the most appropriate spatial patterns for future land use [42,43]. The quality of land information is one of the most influential parameters on the quality of land suitability assessment and the reliability of land use plans [44]. There is a data-driven approach that is commonly used to assess land suitability. A data-driven approach is a quantitative approach based on the relationship between dependent variables (the suitability of seven land cover) and independent variables (land suitability parameters). The data-driven approach that can be used is a supervised probabilistic approach, namely machine learning. In recent decades, machine learning algorithms have been widely used in various land classification cases. The typical machine learning algorithms are artificial neural networks (ANNs), k-nearest neighbors (kNNs), decision trees (DTs), support vector machines (SVMs), and random forest (RF) [17,65–70].

SVM is one of the highest accuracy methods for land classification [17,66]. SVM is used to find the best hyperplane by maximizing the distance between classes. A hyperplane is a function that can be used to divide between classes. Functions used for classification between classes in 2D are referred to as a line, and 3D ones are called planes. In comparison, the functions used for classification within the higher dimensional class are called hyperplanes. Sampling with the Slovin approach was carried out before classification. In the case of land suitability in Java, there are 159,757 grids, so the sample taken is 399 grids. Sample data are represented in $\{(x_1, y_1), \dots (x_i, y_i) \dots, (x_{399}, y_{399})\}$, where $x_i$ represents sample input and $y_i$ represents sample output (land-use/land cover type, i.e., forest). Classifying unclassified data is vital to find a function (x) that is maximally close to $y_i$. Moreover, it can later be generalized to find land suitability. The function for entering sensitivity to bias (noise) is defined first as a variable used to fit the model's accuracy. This function is called the insensitive loss $\varepsilon$, which uses the linear regression function $f(x) = \omega \cdot x + b$:

$$f(x) = \sum_{i=1}^{m} (\alpha_i^* - \alpha_i)(x_i \cdot x) + b \qquad (2)$$

$\alpha_i^* - \alpha_i$ a is obtained from the quadratic programming method based on optimization theory, where $\alpha_i^* - \alpha_i \neq 0$ is the sample in SVM. The constant b is determined based on the Karush–Kuhn–Tucker (KKT) condition of the quadratic convex polygon programming. This land suitability classification uses a kernel trick because the data samples are non-linear. This kernel function is used to project a non-linear function $\varphi$ into a high dimensional space and then form an optimal class separator (hyperplane). The number of land suitability parameters determines the spatial dimension. The kernel function refers to the existence

of the function $K(x, x') = (\varphi(x). \varphi(x'))$ in the sample input function. After a non-linear transformation is performed, the $x_i. x_j$ from Equation (2) is replaced by $K(x_i. x_j)$.

## 3. Results

### 3.1. Spatial Allocation Based on Physiological Needs Using Ecological Footprint (EF)

The allocation of land cover/land use in Java is calculated based on the population's basic needs. The ecosystem provides these needs through various land-use/land cover types, namely, forest, wetland agriculture, dryland agriculture, plantations, built-up land, pastureland, and inland fish grounds (Table 2). The land-use/land cover area required to meet basic human needs is calculated using an ecological footprint approach in each sector [61,71]. Sectors that are taken into account include food, clothing/textiles, infrastructure, and energy. The land requirement per person in one year is calculated by Equation (1). Table 3 shows the yield factor (YF) and equivalent area factor (EQF) values [64]. Furthermore, the needs per sector are obtained by multiplying the needs per person/capita by the total population of Java Island.

- Indonesia's food sector is grouped into eight categories: grains, tubers, animal food, oils and fats, oily fruits/seeds, nuts, sugar, and vegetables and fruit [72,73]. The food sector is produced from wetland agriculture, dryland agriculture, and plantations. The calculation results of the ecological footprint per person for the food sector can be seen in Table 4.
- The clothing/textile sector is produced with raw materials from natural fibers and synthetic fibers [74,75]. The raw material for textiles in Indonesia, which uses natural fibers (cotton), reaches 42%, and the rest is produced from synthetic fibers [76,77]. Therefore, the raw material for clothing/textiles taken into account in this model is cotton made from plantation land. The calculation results of the ecological footprint per person for the clothing/textile sector can be seen in Table 5.
- The infrastructure sector includes residents' needs for housing and public spaces classified into built-up land types. Calculation of the required built-up land area uses the standard of space requirements per person [78,79]. Infrastructures that require wood include infrastructure with a physical structure, namely, a residence (house), cultural and recreational facilities, shopping and commercial centers, religious facilities, health facilities, and educational facilities. Therefore, the proportion of wood demand for buildings must be considered in this model ($m^3$ of wood/$m^2$ of buildings). Wood as a building material is produced from forest land. The calculation results of the ecological footprint per person for the infrastructure sector can be seen in Table 6.
- The energy sector involved in modeling includes electricity, gas, and fuel oil. The amount of energy needed per person is the average of the total energy use in Java. Energy use data are obtained from the Electricity Statistics provided by the Ministry of Energy and Mineral Resources of the Republic of Indonesia. The energy sector is produced from built-up land and pastureland. The calculation results of the ecological footprint per person for the energy sector can be seen in Table 7.

Spatial allocation is generated by the multiplication of ecological footprint per person and population. The total population of Java Island was 171,829,900 [80]. The unit area used is hectares (ha). Spatial allocation as demand should be compared with land supply. The differences show that several land-use/land cover types have deficits and surpluses (Table 8). The surplus shows the condition that the area of land supply is greater than the land demand.

Meanwhile, the deficit indicates that the area of land supply is smaller than the land demand. Those still in deficit include wetland agriculture, dryland agriculture, plantations, and pastureland. Land-use/land cover experiencing a deficit can be met by changing the surplus land-cover/land. However, land-use/land cover changes must pay attention to the suitability of the land.

**Table 2.** The basic needs and land-use/land cover types required for production.

| Product | | Land-Use/Land Cover Type(s) | |
|---|---|---|---|
| | | **Global Footprint Network** | **KLHK *** |
| Food | Rice and other grains | Cropland | Wetland agriculture and dryland agriculture |
| | Tubers | | Dryland agriculture |
| | Nuts and legumes | | |
| | Vegetables and fruit | | |
| | Sugar | | Plantation |
| | Oil and fat | | |
| | Oily fruit/seeds | | |
| | Meat, fish, poultry, eggs | Grazing land and inland fishing grounds | Pastureland and inland fishing grounds |
| Clothing | Cotton | Cropland | Plantation |
| Infrastructure | Housing | Infrastructure and forest | Built-up land and forest |
| | Public space | | |
| Energy | Electricity | Forest | Forest |
| | Gas fuel | | |
| | Fuel oil | | |

*\* Kementerian Lingkungan Hidup dan Kehutanan* (Ministry of Environment and Forestry, Indonesia).

**Table 3.** The yield factor (YF) and equivalent area factor (EQF) values for the Indonesian region (Global Footprint Network, 2018).

| Land-Use/Land Cover Type(s) | | Factor | |
|---|---|---|---|
| **Global Footprint Network** | **KLHK** | **YF (wm$^2$/m$^2$)** | **EQF (gm$^2$/wm$^2$)** |
| Cropland | Wetland agriculture | 0.98551 | 2.493307631 |
| | Dryland agriculture | 0.98551 | 2.493307631 |
| | Plantation | 0.98551 | 2.493307631 |
| Forest | Forest | 0.61317 | 1.275881855 |
| Grazing land | Pastureland | 2.79968 | 0.458242686 |
| Infrastructure | Built-up land | 0.98551 | 2.493307631 |
| Inland fishing grounds | Inland fishing grounds | 1 | 0.368610417 |

### 3.2. Land Suitability Classification Using SVM

Fulfillment of land-use/land cover allocation that is still in a deficit is carried out by considering the land suitability. Land-use/land cover whose area allocation is smaller than the available area (surplus) includes forest, built land, and inland fishing grounds. Meanwhile, the four other types of land-use/land cover experience a deficit. The spatial allocation is based on the nine physical characteristics of the land in each grid. The four types of land-use/land cover requiring additional locations cannot be converted into other types. This condition is also applied to conservation/protected areas. The grids of surplus land-use/land cover, suitable for the deficit land-use/land cover, are candidates for additional locations.

Figure 7 shows the comparison of land-use/land-cover patterns in the existing conditions with the modeling results. Visually, the model had the same pattern as the existing land-use/land cover in 2016. Wetland agriculture and dryland agriculture areas dominate Java. This fact is following the physical characteristics of Java Island. The region conquered by wetland agriculture is a fluvial plain with alluvium as its constituent material. Allu-

vium material can form potential aquifers with flat morphology support to have abundant water availability throughout the year and fertile soils. Dryland agriculture-dominated areas are structural plains and hills. The dominant soil types are latosol and podzolic with deep solum and low to moderate fertility. Therefore, this area is suitable for farming and raising livestock. The highest probability on each grid shows the logical suitability of land-use/land cover based on the visual analysis.

**Table 4.** Ecological footprint or land demand per person of the food sector in Java Island.

| Food Sector | Needs per Person | | | | | | |
|---|---|---|---|---|---|---|---|
| | Kkal/day | Kkal/capita | Wetland Agriculture * | Dryland Agriculture * | Plantation * | Pasture-Land * | Inland Fishing Grounds * |
| Rice and other grains | 1264.86 | 461,675.03 | 345.766 | | 0.000 | 0.000 | 0.000 |
| Tubers | 328.53 | 119,913.78 | 0.000 | | 0.000 | 0.000 | 0.000 |
| Meat, fish, poultry, eggs | 281.33 | 102,686.45 | 0.000 | 285.328 | 0.000 | 3.083 | 4.202 |
| Nuts and legumes | 65.69 | 23,978.07 | 0.000 | | 0.000 | 0.000 | 0.000 |
| Vegetables and fruit | 104.02 | 37,966.72 | 0.000 | | 0.000 | 0.000 | 0.000 |
| Oil and fat | 117.88 | 43,026.90 | 0.000 | 0.000 | | 0.000 | 0.000 |
| Oily fruit/seeds | 34.84 | 12,716.34 | 0.000 | 0.000 | 51.131 | 0.000 | 0.000 |
| Sugar | 191.69 | 69,966.31 | 0.000 | 0.000 | | 0.000 | 0.000 |

* Unit area in square meters ($m^2$).

**Table 5.** Ecological footprint or land demand per person of clothing/textile sector in Java Island.

| Clothing/Textile Sector | Needs per Person | |
|---|---|---|
| | kg/capita | Plantation * |
| Cotton | 7.5 | 121.039604 |

* Unit area in square meters ($m^2$).

**Table 6.** Ecological footprint or land demand per person of the infrastructure sector in Java Island.

| Infrastructure Sector | Needs per Person | | |
|---|---|---|---|
| | Per Capita | Forest * | Built-up Land * |
| Housing and public space | 34,781 $m^2$ | 0.000 | 28.755 |
| Wood demand | 0.214 $m^3$ | 0.321 | 0.000 |

* Unit area in square meters ($m^2$).

**Table 7.** Ecological footprint or land demand per person of the energy sector in Java Island.

| Energy Sector | Needs per Person | | | |
|---|---|---|---|---|
| | per day | per Capita | Built-up Land * | Pasture Land * |
| Electricity | 2.785 kWh | 1016.52 kWh | | |
| Gas fuel | 0.399 kg | 145.63 kg | 2.344 | 0.000 |
| Fuel oil (household) | 0.0185 lt | 6.7525 lt | | |
| Fuel oil (transportation) | 0.499 lt | 182.135 lt | | |

* Unit area in square meters ($m^2$).

**Table 8.** Spatial allocation (ha) based on physiological needs and the fulfillment in Java Island: (1) forest, (2) wetland agriculture, (3) dryland agriculture, (4) plantation, (5) built-up land, (6) pastureland, and (7) inland fishing grounds.

|  | 1 | 2 | 3 | 4 | 5 | 6 | 7 |
|---|---|---|---|---|---|---|---|
| Food | 0.00 | 5,941,290.02 | 4,902,791.44 | 878,581.02 | 0.00 | 52,972.67 | 72,209.85 |
| Clothing/Textile | 0.00 | 0.00 | 0.00 | 2,079,822.30 | 0.00 | 0.00 | 0.00 |
| Infrastructure | 5,512.92 | 0.00 | 0.00 | 0.00 | 494,096.88 | 0.00 | 0.00 |
| Energy | 0.00 | 0.00 | 0.00 | 0.00 | 40,275.42 | 7.47 | 0.00 |
| Land Demand (ha) | 5,512.92 | 5,941,290.02 | 4,902,791.44 | 2,958,403.33 | 534,372.30 | 52,980.15 | 72,209.85 |
| Land Supply (ha) | 2,157,003.69 | 3,867,820.97 | 4,296,050.04 | 377,052.80 | 1,112,210.77 | 50,635.94 | 162,895.49 |
| Difference (ha) | 2,151,490.77 | −2,073,469.05 | −606,741.40 | −2,581,350.52 | 577,838.47 | −2344.20 | 90,685.65 |

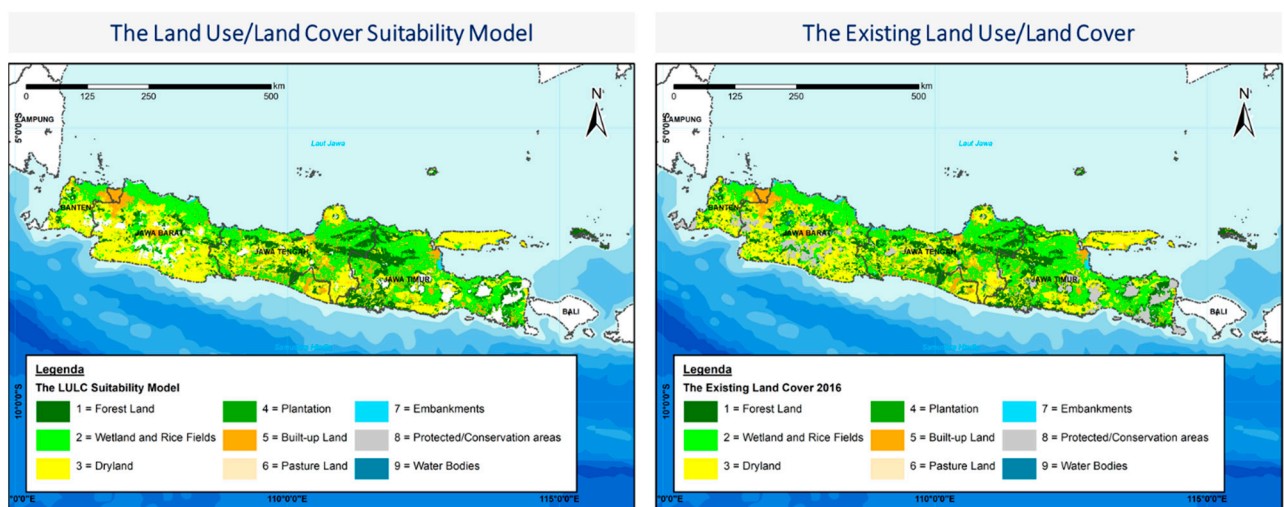

**Figure 7.** Comparison of land suitability model with existing conditions (the year 2016).

The modeling accuracy was obtained from the confusion matrix between the existing land-use/land cover and the model (Figure 8). The model has the same pattern as the existing land-use/land cover (LULC). Dryland and wetland agriculture is the most extensive land-use/land cover in Java. However, the area from the land suitability model with a deficit is still smaller than the LULC allocation requirement, especially for plantation areas. The site suitable for the plantation area based on the modeling results is only 377,052.80 ha, while the need is 2,581,350.53 ha. The difference in area between the models and the existing shows the land suitability model's accuracy. The overall accuracy of land suitability modeling results is 87.728%. The kappa coefficient value from modeling is 0.812, with a significance value of 0.00. The kappa coefficient value shows a very high agreement between the existing land-use/land cover and the model in assessing the land suitability [81]. The significance value is also smaller than the significance level used of 5% (0.00 < 0.05). Thus, there is a significant agreement between the existing land-use/land cover and the model at the 5% significance level.

| | | LAND SUITABILITY MODEL (HECTARES) | | | | | | | | | TOTAL |
|---|---|---|---|---|---|---|---|---|---|---|---|
| | | 1 | 2 | 3 | 4 | 5 | 6 | 7 | 8 | 9 | |
| LULC 2016 (HECTARES) | 1 | 1,663,664.81 | 30,943.12 | 436,862.14 | 18,986.14 | 4,726.81 | 1,015.94 | 804.73 | | | 2,157,003.69 |
| | 2 | 30,293.87 | 3,630,322.64 | 94,556.33 | 1,420.53 | 104,362.95 | 190.57 | 6,674.08 | | | 3,867,820.97 |
| | 3 | 268,667.60 | 88,976.97 | 3,827,830.07 | 23,059.43 | 85,447.98 | 1,672.22 | 395.77 | | | 4,296,050.04 |
| | 4 | 32,683.57 | 4,319.02 | 167,938.12 | 170,380.44 | 1,018.41 | 677.01 | 35.43 | | | 377,052.80 |
| | 5 | 9,133.43 | 52,591.41 | 88,454.53 | 928.82 | 959,171.51 | 679.64 | 1,251.44 | | | 1,112,210.77 |
| | 6 | 11,291.95 | 720.60 | 10,789.77 | 1,265.43 | 2,084.34 | 24,332.57 | 151.28 | | | 50,635.94 |
| | 7 | 837.57 | 41,600.25 | 1,182.97 | 0.00 | 2,333.11 | 0.00 | 116,941.59 | | | 162,895.49 |
| | 8 | | | | | | | | 1,198,021.13 | | 1,198,021.13 |
| | 9 | | | | | | | | | 43,259.79 | 43,259.79 |
| TOTAL | | 2,016,572.80 | 3,849,474.01 | 4,627,613.92 | 216,040.78 | 1,159,145.51 | 28,568.35 | 126,254.32 | 1,198,021.13 | 43,259.79 | 13,264,950.62 |

(**a**)

| | | LAND SUITABILITY MODEL (%) | | | | | | | | | TOTAL |
|---|---|---|---|---|---|---|---|---|---|---|---|
| | | 1 | 2 | 3 | 4 | 5 | 6 | 7 | 8 | 9 | |
| LULC 2016 (%) | 1 | 12.542% | 0.233% | 3.293% | 0.143% | 0.036% | 0.008% | 0.006% | | | 16.261% |
| | 2 | 0.228% | 27.368% | 0.713% | 0.011% | 0.787% | 0.001% | 0.050% | | | 29.158% |
| | 3 | 2.025% | 0.671% | 28.857% | 0.174% | 0.644% | 0.013% | 0.003% | | | 32.386% |
| | 4 | 0.246% | 0.033% | 1.266% | 1.284% | 0.008% | 0.005% | 0.000% | | | 2.842% |
| | 5 | 0.069% | 0.396% | 0.667% | 0.007% | 7.231% | 0.005% | 0.009% | | | 8.385% |
| | 6 | 0.085% | 0.005% | 0.081% | 0.010% | 0.016% | 0.183% | 0.001% | | | 0.382% |
| | 7 | 0.006% | 0.314% | 0.009% | 0.000% | 0.018% | 0.000% | 0.882% | | | 1.228% |
| | 8 | | | | | | | | 9.031% | | 9.031% |
| | 9 | | | | | | | | | 0.326% | 0.326% |
| TOTAL | | 15.202% | 29.020% | 34.886% | 1.629% | 8.738% | 0.215% | 0.952% | 9.031% | 0.326% | 100% |

(**b**)

**Figure 8.** The confusion matrix of the existing land-use/land cover (LULC) and land suitability model: (**a**) land suitability model (hectares) and (**b**) land suitability model (%).

## 4. Discussion

### 4.1. The Overall Performance of Land Suitability Model in Java Island

Thematic accuracy is one of the elements of geospatial data quality [82]. Thematic accuracy is defined as the accuracy of quantitative attributes and non-quantitative attributes in feature classification and the relationship between features. There are many different ways to look at the thematic accuracy of classification. The confusion matrix allows the calculation of some accuracy metrics, i.e., accuracy, precision, recall, and F1-score. These indicators are used to assess model performance objectively. Classifier, in this case, SVM, can predict which land is suitable (true-positive) and not suitable (true-negative) for a particular LULC type. The SVM classifier can also make mistakes or errors when predicting suitable (false-positive) and unsuitable (false-negative) locations for certain LULC types. Confusion matrix components in the form of true-positive (TP), true-negative (TN), false-positive (FP), and false-negative (FN) can be used to calculate thematic accuracy indicators.

Table 9 shows the value indicators or standard measures to assess the land suitability model's performance in Java Island. The indicators were the accuracy, precision, recall, specificity, and F1-score calculated for the seven LULC types whose suitability is predicted. The seven types of LULC calculation results are then averaged to measure the land suitability model's average macro performance. The highest accuracy value is owned pasturelands by an almost perfect value, which is 99.75%. Accuracy indicates the model's ability to predict land suitability and unsuitability correctly. LULC type with high accuracy does not mean the precision and recall values will be automatically high too. For example, forest land has an accuracy of 92.97%, but the precision and recall values are only 82.46% and 77.26%. This condition is inversely proportional to dryland agriculture, where the accuracy is only 89.50%, but the precision and recall values are more significant than the forest land. The value of precision shows the consistency level of the classification determined by comparing the classification results with the conditions in the field. In comparison, recall shows prediction correctness level for all identifiable objects. The precision and recall for any given LULC type typically are not the same. In Table 9, the plantations' precision was 97.90%, while the recall was 77.87%. This fact means that even though 97.90% of the reference plantations areas have been correctly predicted as suitable for 'plantations',

only 77.87% percent of the areas predicted suitable for "plantations" in the classification were actually suitable for plantations.

**Table 9.** The model performance of the land suitability model in Java Island using SVM.

| Code | LULC Type | Accuracy | Precision | Recall | Specificity | F1-Score |
|------|-----------|----------|-----------|--------|-------------|----------|
| 1 | Forest | 92.97% | 82.46% | 77.26% | 96.41% | 79.78% |
| 2 | Wetland agriculture | 96.20% | 94.26% | 93.92% | 97.29% | 94.09% |
| 3 | Dryland agriculture | 89.50% | 82.92% | 88.94% | 89.82% | 85.82% |
| 4 | Plantations | 97.90% | 77.87% | 46.32% | 99.57% | 58.09% |
| 5 | Built-up land | 97.06% | 82.76% | 86.14% | 98.17% | 84.41% |
| 6 | Pastureland | 99.75% | 82.09% | 51.28% | 99.95% | 63.12% |
| 7 | Inland fishing grounds | 99.54% | 92.03% | 72.11% | 99.91% | 80.86% |
| | MACRO (Average) | 96.13% | 84.91% | 73.71% | 97.30% | 78.02% |

Figure 9 shows the distribution patterns of accuracy, precision, and recall values for the seven types of LULC. Visually, the accuracy value distribution pattern has the most stable pattern, with values always above 90%. Meanwhile, the precision and recall values have a similar pattern. The pattern of recall values fluctuated, especially in plantation and pastureland. This condition causes macro-precision and macro-recall values to be below 90%. That pattern shows there might be a class imbalance. Moreover, the F1-scores were calculated to assess model performance. The F1-score is an average of precision and recall by weight. Therefore, this score considers both false positives and false negatives. F1 is typically more useful than accuracy, especially if there is an imbalanced class distribution. Macro-F1 of the model shows an increase in macro-recall value, but it is still below 80%. Therefore, it is necessary to calculate the average micro performance, such as the overall accuracy value.

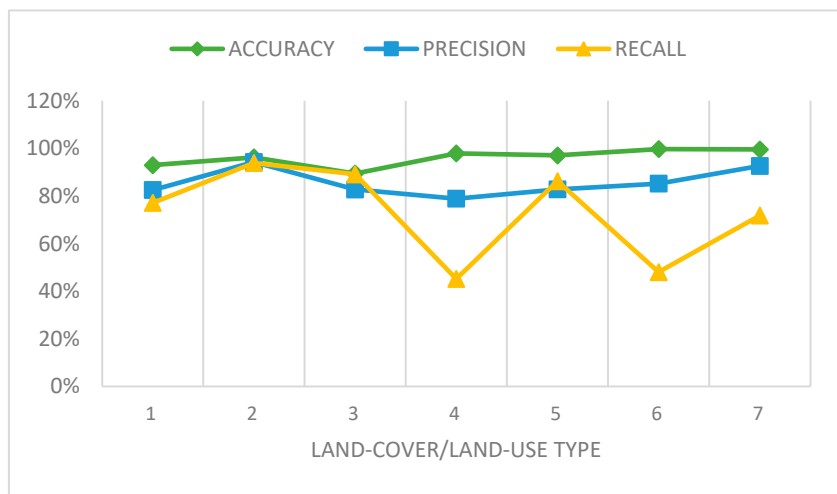

**Figure 9.** The plot of accuracy, precision, and recall values based on the average of seven land-use/land cover types.

The micro-average performance aggregates the contributions of all land-use/land cover types to compute the average metric. An overall accuracy is a form of micro average performance. The precision, recall, and F1-score indicators on the micro average performance will always have the same value as the overall accuracy (Table 10). Overall accuracy is essentially taken from of all of the reference sites whose proportions were mapped correctly. The diagonal elements on the confusion matrix represent the areas that were correctly classified. Overall accuracy was calculated by adding the number of correctly

classified sites and dividing it by the total number of reference sites. The calculation results show the overall accuracy of the land suitability model is 86.46%. This fact means that the model's error percentage reaches 13.54% (100% − overall accuracy). The model also has a kappa value of 0.812 or equivalent to a very high level of agreement between the model and existing conditions.

**Table 10.** The overall accuracy of the land suitability model in Java Island using SVM.

| Overall Accuracy | 86.46% |
|:---:|:---:|
| Micro-F1 | 86.46% |
| Micro-Precision | 86.46% |
| Micro-Recall | 86.46% |

The land suitability model's overall performance, summarized over all possible thresholds, is given by the receiver operating characteristics (ROC) curve. The name "ROC" is historical and comes from communications theory. ROC curves are used to see the classifier's ability, in this case, SVM, to separate the positive and negative classes and identify the best threshold for separating them. The ROC curve is plotted using a false positive rate (FPR) as the X-axis obtained from 1 – specificity and a true positive rate (TPR) data or sensitivity as the Y-axis. Specificity has the same definition as the recall value. The model's performance is considered inferior if the resulting curve approaches the baseline or a line that crosses from point 0.0 and good if the curve approaches 0.1. Figure 10 shows a blue curve that shows promising performance because the TPR value continues to approach 0.1 for the FPR value from zero to 76.20%. In addition, the calculation of the area under the curve (AUC) is also carried out. An excellent model has AUC near 1.0, which means it has a good separability measure. The land suitability model's AUC value reaches 0.842, which can be classified as an excellent model.

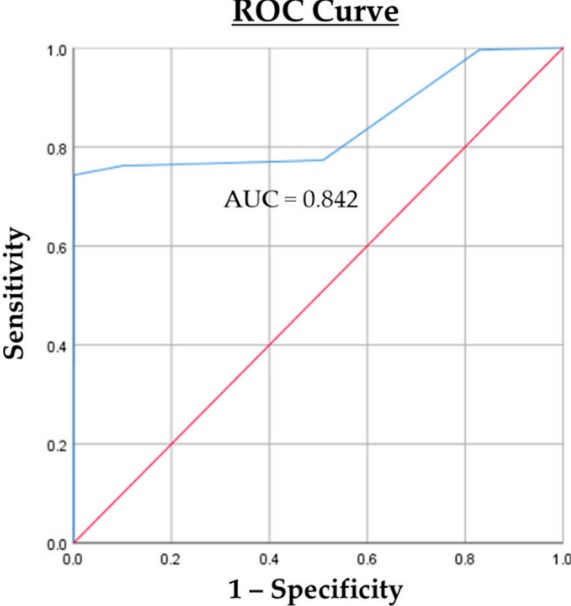

**Figure 10.** The receiver operating characteristics (ROC) curve and area under the curve (AUC) of the land suitability model overall performance.

## 4.2. Fulfillment of Spatial Allocation Based on Land Suitability

The land requirements (demand) can be met using the available land-use/land cover (supply). The comparison between the supply and demand of land-use/land cover (LULC)

in Java Island shows that four LULC classes experience deficits. LULC classes that have deficits are wetland agriculture, dryland agriculture, plantations, and pastures. Meanwhile, LULC classes that experience a surplus are forest, built-up land, and inland fishing grounds (Table 11). These results have the same pattern as the ecological footprint study conducted by the Ministry of Public Works of the Republic of Indonesia (2010) and Nathaniel (2020) [83,84]. LULC classes that have a deficit must be met to meet the needs of the population in Java.

**Table 11.** Allocation fulfillment of land-use/land cover in Java Island based on the land suitability.

| Code | LULC Type | Supply (hectare) | Demand (hectare) | Difference (hectare) |
|------|-----------|------------------|------------------|----------------------|
| 1 | Forest | 2,157,003.69 | 5512.92 | 2,151,490.77 |
| 2 | Wetland agriculture | 3,867,820.97 | 5,941,290.02 | −2,073,469.05 |
| 3 | Dryland agriculture | 4,296,050.04 | 4,902,791.44 | −606,741.40 |
| 4 | Plantations | 377,052.80 | 2,958,403.33 | −2,581,350.53 |
| 5 | Built-up land | 1,112,210.77 | 534,372.30 | 577,838.47 |
| 6 | Pastureland | 50,635.94 | 52,980.15 | −2344.21 |
| 7 | Inland fishing grounds | 162,895.49 | 72,209.85 | 90,685.64 |
| 8 | Conservation/Protected Area | 1,198,021.13 | 1,198,021.13 | 0.00 |
| 9 | Water body | 43,259.79 | 43,259.79 | 0.00 |
| | Total | 13,221,690.83 | 15,665,581.14 | −2,443,890.31 |

Land suitability modeling results can be used to find new candidate locations for the deficit land-use/land cover. The land-use/land cover (LULC) experiencing a deficit should not be converted into another type of land-use/land cover (Figure 11). This constraint also applies to built-up land because it is mostly permanent. The modeling results show that all land-use/land cover types cannot be changed to have a suitable location for the deficit LULC, which is 8,191,106.50 ha. The rest is in the form of forest land and inland fishing grounds with 530,590.57 ha. The total area is still smaller than the difference between land supply and land demand, namely, forest 487,807.35 ha < 2,151,490.77 ha and inland fishing grounds 42,783.22 ha < 90,685.64 ha (Figure 11 and Table 11). This condition must be ensured to avoid increasing the deficit status in other types of LULC. Therefore, the grids are available for conversion to wetland agriculture, dryland agriculture, plantations, and pastureland. Forestland contributes the most to the changeable grid because of its dominant areas on Java Island. Forestland classification in this research does not include conservation/protected areas. The forest land classification consists of secondary dryland forests, plantation forests, and shrubs.

The candidate grid can be used to calculate the area of possible land-use/land cover (LULC) allocation fulfillment (Table 12). The land suitability in Java Island is dominated by agricultural land consisting of wetland agriculture, dryland agriculture, and plantations. Possible fulfillment is obtained with the land supply formula minus changeable LULC, then added with the candidate grid area. Change-able LULC is a grid(s) of the surplus LULC type, which is available to be converted into the deficit LULC type. Dryland agriculture has the most extensive candidate grids covering 438,045.11 ha or equivalent to 5111 grids. Meanwhile, pasturelands have candidate grids of at least 1015.94 ha or equivalent to 23 grids. The modeling results show that all deficit LULC types have new candidate locations, but have not been able to change the deficit status. LULC change based on the possible allocation fulfillment calculation will also reduce the ecosystem's negative impact. This benefit is because LULC change will be based on land suitability.

The difference between land supply with land demand and the possible fulfillment based on its suitability shows a reduction in the deficit areas (Table 13). The number of LULC types experiencing deficit is still the same after adding the candidate grids. The area

of forest is reduced by 22.62%, which is spread over four deficit LULC types. Candidates from forest land are dominated by dryland agriculture, covering 436,862.14 ha (89.56% of changeable forest land areas). In comparison, the inland fishing grounds were reduced by 42,783.22 ha or 26.26%, which was distributed to fulfill the allocation of wetland agriculture and dryland agriculture. The total area of the initial deficit based on available land (supply) was 5,263,905.19 ha, which was dominated by cropland areas. The deficit reduction reached 4,733,314.62 ha with the same pattern. Meeting the needs with this scenario is acceptable because it had succeeded in reducing the deficit area and not adding to the LULC types with a deficit status.

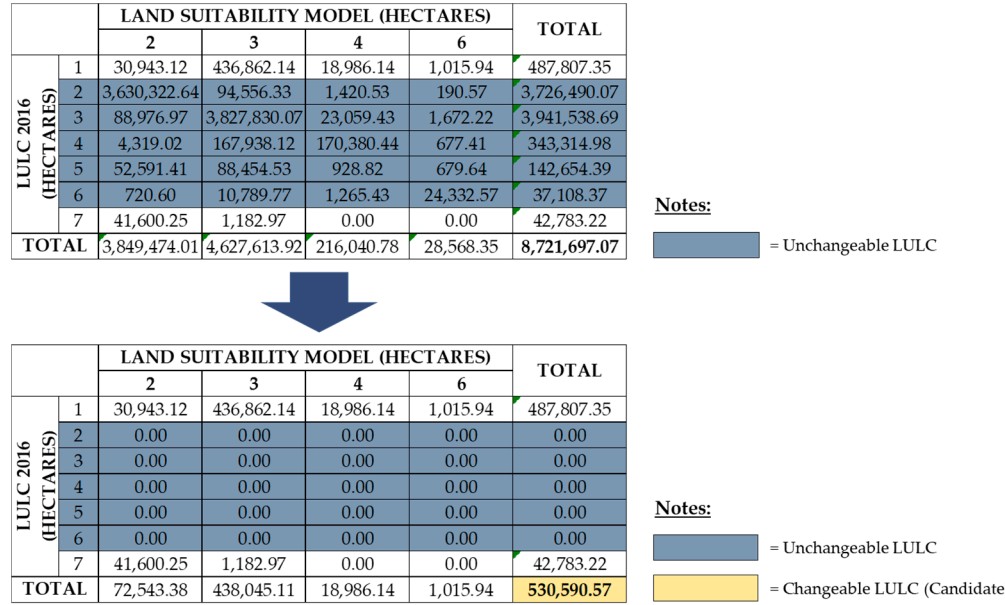

**Figure 11.** The total area of changeable or possible candidate of the deficit land-use/land cover(s) (LULC).

**Table 12.** The possible allocation fulfillment of land-use/land cover in Java Island based on the land suitability.

| LULC Type | Supply * | Changeable * | Candidate * | Possible Fulfillment * |
|---|---|---|---|---|
| Forest | 2,157,003.69 | 487,807.35 | 0.00 | 1,669,196.35 |
| Wetland agriculture | 3,867,820.97 | 0.00 | 72,543.38 | 3,940,364.35 |
| Dryland agriculture | 4,296,050.04 | 0.00 | 438,045.11 | 4,734,095.15 |
| Plantations | 377,052.80 | 0.00 | 18,986.14 | 396,038.94 |
| Built-up land | 1,112,210.77 | 0.00 | 0.00 | 1,112,210.77 |
| Pastureland | 50,635.94 | 0.00 | 1015.94 | 51,651.88 |
| Inland fishing grounds | 162,895.49 | 42,783.22 | 0.00 | 120,112.27 |
| Conservation/Protected Area | 1,198,021.13 | 0.00 | 0.00 | 1,198,021.13 |
| Water body | 43,259.79 | 0.00 | 0.00 | 43,259.79 |
| **Total** | **13,221,690.83** | **530,590.57** | **530,590.57** | **13,221,690.83** |

* Unit area in hectares (ha).

There are several solutions to overcome the deficit of land-use/land cover (LULC). The first solution, namely all commodities from land-use/land cover that are experiencing deficit are filled with imports from other islands or abroad. However, this will only cause new problems, especially for the country's economic condition. Indonesia's dependence on imports is one reason for weakening the rupiah exchange rate against the United States (US)

dollar [84]. Therefore, high import figures can lead to inflation. Meanwhile, Indonesia's import figures are currently high, although they have decreased compared with 2018 [85]. The second solution is the population transmigration program from Java Island to other islands in Indonesia. However, the transmigration program in the current development era creates new problems. Newcomers or transmigrants are growing faster than local communities. This problem has resulted in social gaps in local communities. The third solution, namely, optimization of the existing land-use/land cover, will be imported from other islands or abroad. This last solution is the best compared with the other solution options. Optimization of land-use/land cover location-allocation can be done by fulfilling the LULC allocations that are still in a deficit using the surplus LULC. For example, deficit PL needs such as wetland agriculture, dryland agriculture, and plantations can be partially met by allocating forest land that is still in a surplus. The requirements that must be met for finding the location of LULC that are in a deficit are the land suitability. Each location is theoretically compatible with multiple LULC types, but with different compatibility levels. The PL suitability level can be represented in the posterior probability value. Therefore, it is necessary to model the suitability of PL with the posterior probability value at the next stage.

**Table 13.** The comparison of supply based fulfillment and land suitability based fulfillment in Java Island.

| Code | LULC Type | Supply (ha) | Demand (ha) | Difference (ha) | LSM * (ha) | Difference (ha) |
|---|---|---|---|---|---|---|
| 1 | Forest | 2,157,003.69 | 5512.92 | 2,151,490.77 | 1,669,196.35 | 1,663,683.43 |
| 2 | Wetland agriculture | 3,867,820.97 | 5,941,290.02 | −2,073,469.05 | 3,940,364.35 | −2,000,925.67 |
| 3 | Dryland agriculture | 4,296,050.04 | 4,902,791.44 | -606,741.40 | 4,734,095.15 | −168,696.29 |
| 4 | Plantations | 377,052.80 | 2,958,403.33 | −2,581,350.53 | 396,038.94 | −2,562,364.39 |
| 5 | Built-up land | 1,112,210.77 | 534,372.30 | 577,838.47 | 1,112,210.77 | 577,838.47 |
| 6 | Pastureland | 50,635.94 | 52,980.15 | −2344.21 | 51,651.88 | −1328.27 |
| 7 | Inland fishing grounds | 162,895.49 | 72,209.85 | 90,685.64 | 120,112.27 | 47,902.42 |
| 8 | Conservation/Protected Area | 1,198,021.13 | 1,198,021.13 | 0.00 | 1,198,021.13 | 0.00 |
| 9 | Water body | 43,259.79 | 43,259.79 | 0.00 | 43,259.79 | 0.00 |
| | **TOTAL** | **13,221,690.83** | **15,665,581.14** | **−2,443,890.31** | **13,221,690.83** | **−2,443,890.31** |
| | | | DEFICIT | **−5,263,905.19** | | **−4,733,314.62** |

\* Land Suitability Model.

Table 14 shows the comparisons of various land requirement calculations using the ecological footprint approach. Land requirements were calculated based on physiological needs in 2016, except for calculations by the Ministry of Public Works in 2010. It can be seen that the calculation results of the three studies have significant differences. This condition is because GFN calculates land requirements with derivatives from global standards (world-average productivity). Meanwhile, the calculations made by the Ministry of Public Works and Safitri et al. use national standards. The Ministry of Public Works also takes into account the number of imports and exports of goods. Therefore, the Ministry of Public Works' calculation of land requirements is smaller than that of Safitri et al. The land requirement for cropland in the three studies has the highest value. The cropland consists of agriculture wetland, agriculture wetland, and plantation.

**Table 14.** The comparison of land requirements using the ecological footprint approach in this study with other studies.

|  | Global Footprint Network (gha) | The Ministry of Public Works (ha) | Safitri et al. (ha) |
|---|---|---|---|
| Cropland | 135,752,024.20 | 4,343,805.00 | 13,802,484.79 |
| Pastureland | 4,857,696.85 | 1715.00 | 52,980.15 |
| Forest | 69,742,317.43 | 12,616.00 | 5512.92 |
| Fishing ground | 68,405,182.80 | 2,047,015.00 | 72,209.85 |
| Built-up land | 17,612,614.72 | 130,933.00 | 534,372.30 |

## 5. Conclusions

Spatial allocation based on physiological needs is very challenging because of complex variables, depending on the study area. Physiological needs are very dependent on the social, economic, and environmental conditions of the study area. Therefore, it is necessary to have clear boundaries in determining the groups of needs involved. This research uses four sectors of needs with regional coverage of Java Island. The sector classification of needs will be different and more detailed when used for smaller areas (i.e., districts/cities). In this study, the land requirements to absorb emissions produced by each sector have not been involved. The calculation of needs also uses national standards; in fact, each region with various socio-economic conditions will have local standards. Hence, the ecological footprint approach is proven to measure land needs based on consumption patterns and environmental capacity to supply resources. As a tool, the ecological footprint's accuracy depends on the quality of the calculation's data.

Furthermore, SVM can produce highly accurate land suitability models (86.46%) with a very high level of agreement between the model and existing conditions (0.842). The model's accuracy in this study is better than the previous studies, which only reached a maximum of 85%. However, this land suitability model only uses physical parameters in the form of land characteristics. In principle, land use planning is successful only if it can be implemented in any local conditions of social, cultural, political, and economical. Therefore, it is necessary to include social, cultural, political, and economic parameters to assess land suitability.

Fulfilling spatial allocation with a combination of ecological footprint and land suitability models has succeeded in reducing the deficit area without changing land status from surplus to deficit. This land suitability model is proven to be able to reduce the land deficit by up to 530,590.57 ha or equivalent to 10.08% of the initial deficit. It should be noted that the spatial allocation in this study only considers land suitability. Furthermore, it is also necessary to develop a location-allocation analysis to overcome land-use/land cover patterns' inefficiency. This condition causes various social, economic, and environmental problems. This development is in line with the theory of sustainable development.

**Author Contributions:** Conceptualization, Methodology, Validation, Formal Analysis, Funding Acquisition, Sitarani Safitri; Software, Investigation, Sitarani Safitri, Akhmad Riqqi, Albertus Deliar, and Irawan Sumarto; Resources, Irawan Sumarto; Data Curation, Sitarani Safitri and Akhmad Riqqi; Writing—Original Draft Preparation, Sitarani Safitri; Writing—Review & Editing, Sitarani Safitri, Akhmad Riqqi, and Albertus Deliar; Visualization, Sitarani Safitri; Supervision, Ketut Wikantika; Project Administration, Irawan Sumarto. All authors have read and agreed to the published version of the manuscript.

**Funding:** This research and publication of this article was funded by Lembaga Pendidikan Lembaga Pengelola Dana Pendidikan Indonesia.

**Data Availability Statement:** This research using publicly available datasets. This data can be found here: https://data.isric.org/geonetwork/srv/eng/catalog.search (accessed on 10 March 2021) and https://earthexplorer.usgs.gov/ (accessed on 10 March 2021).

**Acknowledgments:** In addition, the authors would like to acknowledge the support from the Institute for Research and Community Service (LPPM), Bandung Institute of Technology through Research Program, Community Service and Innovation—*Program Penelitian, Pengabdian kepada Masyarakat dan Inovasi* (P3MI) ITB 2020.

**Conflicts of Interest:** The authors declare no conflict of interest. The funders had no role in the design of the study; in the collection, analyses, or interpretation of data; in the writing of the manuscript; or in the decision to publish the results.

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
