# Peer review of "Spatial Allocation Based on Physiological Needs and Land Suitability Using the Combination of Ecological Footprint and SVM (Case Study: Java Island, Indonesia)"

_ijgi, doi:10.3390/ijgi10040259_

Round 1

Reviewer 1 Report

I suggest a revision: 1. Draw a flowchart from your work flow that briefly shows the process and in the Discussion section. 2. Compare your results with the results of other researchers. 3. The literature review part may be further improve.  4. In abstract and conclusion, authors need to add some numerical results. 5. Description of the figures should be more complete 6. Improve the English.   7. The conclusion should include more details.  8. Please improve the Conclusion section, this section need to more details and more explain. 9. Please refer to new paper about this field.

Author Response

Dear Reviewer 1,

Here I attached the response to the reviewer's comments. Thank you.

Sincerely,

Sitarani Safitri

Reviewer 2 Report

Contribution

This manuscript describes a study aimed at identifying the most suitable land covers / land uses for achieving sustainable development on Java Island, Indonesia. To model land suitability, the authors used the ecological footprint concept and the support vector machine method.

Significance

  • The theoretical and methodological contributions of the research are unclear. A stronger manuscript would articulate how the research advances the fields of land system science, sustainability science, or spatial science, for example. It would also describe any advances in methods or creative ways in which existing methods were used.
  • Findings from the research, if deemed accurate, may inform land use planning on Java Island.

Novelty

  • The introduction of a stronger manuscript would provide a richer review the existing literature and articulate how the present study creatively and significantly advances theory and/or methods.

Concerns

  • The manuscript is mostly comprehensible. However, some statements are unclear and, while the sentences are refreshingly short, they are so short that paragraphs at times appear to be long lists of only loosely connected sentences. As a result, I found it difficult to understand significant portions of the manuscript without re-reading passages multiple times. I think the manuscript needs some serious English language editing.
  • The Introduction notes an overall goal, but no specific objectives, research questions, or hypotheses.
  • The idea of land suitability needs to be better defined, both in descriptive and quantitative terms.
  • While described in great detail at times, the methods are unclear. I do not understand from the manuscript how EF and land suitability are connected. Moreover, SVM is a supervised classifier - it is unclear how the classifier was trained or tested.
  • The terms deficits and surpluses are a bit odd - they need to be more clearly defined.
  • "Optimization of land-use/land cover location-allocation" may be an option to address the "LULC deficit", but the discussion in the current manuscript somewhat ignores the reality that there are people on the ground who may or may not be open to this kind of optimization. A discussion that acknowledges social, cultural, political, economic, etc. challenges and potentials in optimizing LULC would add value to the manuscript.

Minor Issues

  • Figure 2 is illegible
  • Table 8: what do 1, 2, 3, etc. represent? --- If LULC, replace all numbers with actual LULC or provide details in captions. Same applies to other figures and tables.
  • It would be better to discuss model performance (accuracy, precision, recall) before all other findings so that readers get a feel for how uncertain those other findings might be.

Author Response

Dear Reviewer 2,

Here I attached the response to the reviewer's comments. Thank you.

Sincerely,

Sitarani Safitri

Reviewer 3 Report

The article is interesting and of high usability, but there are some weaknesses in it that should be improved. First of all, the introduction needs improvement. It lacks essential elements that are necessary in this chapter, i.e. the main objective and possibly specific objectives, the research hypothesis or thesis, and lacks emphasis on the innovative nature of the research. What new contributions do the authors make to world science through their research?

In the abstract, many sentences begin the same eg: "In this study...". This unnecessary repetition creates a feeling of confusion and loss for the reader.

The methodology adopted is well-founded, but as the authors have acknowledged, it has significant shortcomings in that it assumes a limited number of factors to fully examine the phenomenon.

I also have two specific comments:

1) P.3, 107-109. “The purpose of data collection is to obtain the required information …. The data collected in this study is secondary data identified based on the variables in the hypothesis.”

Such obvious phrases, well known to all scientists, do not need to be placed in research papers at a high level of scientific discussion. They unnecessarily lengthen the scientific argument. Instead, one should formulate a research hypothesis and refer to it as a justification for the adopted scope of research.

2) p. 5, 140-147. “Several methods have been developed to calculate….. Therefore, in this study, the calculation of spatial allocation used the EF approach.”

Unnecessary repetition. This paragraph is in the introduction.

I recommend the article for publication after necessary improvement.

Author Response

Dear Reviewer 3,

Here I attached the response to the reviewer's comments. Thank you.

Sincerely,

Sitarani Safitri

Round 2

Reviewer 1 Report

This manuscript is acceptable.